# The Micro Topology and Statistical Analysis of the Forces of Walking and Failure of an ITAP in a Femur

**DOI:** 10.3390/mi12030298

**Published:** 2021-03-12

**Authors:** Euan Langford, Christian Andrew Griffiths, Andrew Rees, Josh Bird

**Affiliations:** 1The Defence Science and Technology Laboratory (DSTL), Fareham PO17 6AD, UK; elangford@dstl.gov.uk; 2College of Engineering, Swansea University, Swansea SA1 8EN, UK; Andrew.Rees@Swansea.ac.uk; 3Department of Mechanical Engineering, University of Bath, Bath BA2 7AY, UK; jb3315@bath.ac.uk

**Keywords:** prostheses, ITAP, micro topology, ANSYS, MATLAB, additive manufacture

## Abstract

This paper studies the forces acting upon the Intraosseous Transcutaneous Amputation Prosthesis, ITAP, that has been designed for use in a quarter amputated femur. To design in a failure feature, utilising a safety notch, which would stop excessive stress, σ, permeating the bone causing damage to the user. To achieve this, the topology of the ITAP was studied using MATLAB and ANSYS models with a wide range of component volumes. The topology analysis identified critical materials and local maximum stresses when modelling the applied loads. This together with additive layer manufacture allows for bespoke prosthetics that can improve patient outcomes. Further research is needed to design a fully functional, failure feature that is operational when extreme loads are applied from any direction. Physical testing is needed for validation of this study. Further research is also recommended on the design so that the σ within the ITAP is less than the yield stress, σ_s_, of bone when other loads are applied from running and other activities.

## 1. Introduction

The failure features of the femur are unique to the human skeleton because the bone is designed to transfer σ throughout the structure when compressive or tensile loads are applied. The σ is also dispersed to the surrounding muscle, cartilage and connected bone in the skeleton. This allows the femur to handle large σ when the user is in motion such as running and jumping. However, the femur is not indestructible as the bone structure is susceptible to fracture from extreme shear loads. It has been theorised by that a sheer force of 4000 N applied perpendicular to the bone would cause a fracture [1]. The properties of the individual’s bone structure are also key factors for the propagation of fracturing as the physiology is unique to the individual. This means that some individuals and ages are more susceptible to fractures particularly when bone decreases after the age of 30 [2]. The femur can also fail under different extreme loads from, for example, collision or devices such as landmines, improvised explosive devices, IEDs, and other explosives as identified by Mckay et al. [3]. The direction and position of the applied σ is difficult to define, because the load could be applied from any direction at any time.

The intraosseous transcutaneous amputation prosthesis, ITAP, is a direct skeleton attachment that is used by amputees [4]. This is used rather than a standard socket interface that causes swelling, discomfort and potential infection for the user. The socket interface often prohibits the life span of the use of the prosthetic [5]. As a result, the development of the ITAP and similar direct skeleton attachments have been designed to mitigate these problems, to improve the quality of life for the user [4]. ITAPs were initially designed from studying animals with protruding bone such as deer and rhinos [6]. The mechanics and biomechanics of these animals’ bone development have been analysed due to the lack of infection of the tissue surrounding the protruding bone which is similar to an embedded prosthetic. This is vital, as an embedded prosthetic can be susceptible to infection and damage of the local tissue and if a prosthetic is not properly manufactured with biomedical procedures with appropriate postproduction methods [4]. Consequently, the ITAP is constructed in a manner that replicates the antlers of deer. An open insertion to the bone with reduced bacterial colonization infection was realised using a diamond-like carbon (DLC) ITAP [6].

The biomechanics of the ITAP and the human body need to be thought of during design, simulation and production as the material must not be harmful to the patient’s body. As such, the main material used is that of Titanium (Ti). This is because Ti is biocompatible with the human body, with a high processability through Computer-aided design (CAD) and additive manufacturing (AM) processes. Ti is also strong enough to handle the loads associated with walking and other movements [7,8]. However, it can deteriorate in to smaller fragments in the body over time due to movement. This is most commonly attributed to hip and knee implants because there is constant friction between the implant and the joint [9]. Additional alternative materials can be used for prosthetics such as Polyetheretherketone (Peek). This can be processed by AM methods and inserted into the bone to help repair and supplement the bone after injury. This is commonly used for American football players for bone repair therapy as Peek is resistant to infection [10]. However, Peek is not suitable for use in a load bearing ITAP as there is poor bonding between the bone and the material. Therefore, it could easily become possible for the ITAP to be partially or fully dislodged from the bone, causing internal damage to the user.

The ITAP can be used for both cosmetic and load bearing prosthetics; the former can range from prosthetic eyes, fingers, noses, ears amongst others [11], while the latter can be used for prosthetic limbs with loadbearing forces acting upon extremities. This includes, but is not limited to, both high and low leg amputations, either above or below the knee. However, the work of Newcombe et al., identified that an ITAP implant was not advisable for an amputation greater than one quarter of the original length of the bone. This was due to the residual σ in the femur housing, known as the anchor, being too great so damage to the cortical bone could occur [12]. Further suggested amendments to the ITAP have been developed to reduces the σ with in both the bone and the ITAP. This includes a safety notch that is positioned outside the user’s congealed tissue. The proposed modification could prevent an extreme load from damaging the user as the ITAP would failure at the safety notch [13].

Other direct skeletal prosthetics are available for use such as Osseointegrated Prostheses for the Rehabilitation of Amputees, OPRA. This system involves screwing into the bone cavity which increases the residual σ all along the anchor. This system works because it increases the user’s hip flexibility and extension in comparison to traditional socket interfaces. However, over time it was recorded that pain was detected due to the build-up of residual σ [14]. The ITAP is not only suited to use in humans but it has had clinical trial in canines. Noel Fitzpatrick et al. studied the procedure for installation and use in four separate canines. The study identified that all subjects’ quality of life improved due to the prosthetic attachment. However, canines 1–3 were euthanized due to metastatic disease spreading throughout the body; it was unclear if this was related to the ITAP [15]. As the ITAP can be fitted for different purposes and species, bespoke parts can be produced. This is extremely useful as the properties of bone vary as well as the length of the remaining bone after amputation, see Figure 1 and Figure 2. Bone shape and modelling can be recorded by using Computed Tomography (CT) scanning. This creates a need for design iterations to model and build an ITAP that suits the individual’s lifestyle. In this process it is essential to design a build that does not over engineer the forces acting upon the leg. Failure to work in this design constraint could cause near fatal damage to the user resulting in further amputation or loss of life. This is due to the failure of the anchor supporting the ITAP and the potential rupturing of arteries and veins surrounding the localized area of the prosthetic. The work of Sullivan et al. concluded that the use of direct skeleton attachments for prosthetics increases the user’s quality of life and the function of the prosthetic limb. Comfort also increased due to the lack of sores developing on the stump. This is because there is no contact between the stump of the limb and the prosthetic sock fitted over the top. However, the reports concluded that aspects of the development of the ITAP needed further advancement because the “trans-femoral amputee” case needs extensive review before it becomes a common clinical procedure for prosthetic rehabilitation [16]. It has been identified by Bird et al., that the initial area of failure and high σ concentration of the ITAP is located at the root of the ITAP [13]. The results of the study identified that a single safety notch was not enough for the ITAP to be suitable for use. This is because the strength of the Ti used for the ITAP is greater than the bone, so the bone would fail before the ITAP, resulting in damage to the user. Thus, it is identified that a comparable σ between bone and the ITAP at the point of insertion is needed for successful development of the product. This is categorized as designing in failure for success.

The benefit of using AM methods of production is that the topology of the ITAP can be adapted to meet the users bespoke requirement. The manufacturing process allows for modification of the internal structure, of the ITAP and each can be printed to have the same properties and micro topology as the bone being amputated, as seen via CT scanning. The first aim of this study is to identify the optimum topology of the ITAP, locating critical material that is integral to the structure of the design when simulated to both walking and extreme loads. The secondary aim is to optimise the design of the ITAP topology so that a controlled failure can be achieved when extreme loads are applied. Thus, producing an ITAP that prevents a further damage to bone when subjected to excessive loads. For both aims the simulation will consider the design of a quarter amputation of a femur with an ITAP embedded into the bone (Figure 3). Both walking and extreme σ studies are considered because the operational σ of the design needs to be evaluated in order to design a safe prosthetic that can be embedded into the bone. To ensure reliable results a multiple modelling software approach is used to generate the topologies. The paper is organised as follows: the next section uses simulation methods to identify the maximum stress, σ_Max_, on the standard ITAP due to walking loads. This is followed by topology optimisation of the ITAP model using ANSYS and MATLAB. Then, Section 4. considers the influence of the topology of the ITAP design when under extreme loads. Finally, conclusions are presented on the optimisation of ITAP designs.

## 2. Experimental Methodology

The purpose of the research is to establish the strength of the ITAP, and to identify the behaviour of a safety mechanism to prevent excessive forces transferring to bone. The focus will be on performing Simulations of σ_Max_ on the ITAP due to walking loads. In order to perform this the following section will identify initial design, boundary conditions and mesh used. In 2.2 the Simulated results of the ITAP standard design will be presented and then used as a benchmark for the ITAP designs with topology optimisation in Section 3.

### 2.1. The Initial Design, Boundary Conditions and Mesh

The A quarter amputation of the femur requires an ITAP with an embedded depth of 160 mm, with a diameter of 14 mm [12]. The femur provided in this study was obtained by CT scanning of a deceased male aged 44 weighing at 85 kg and at a height of 185 cm [17]. Based on this femur the standard ITAP design can be seen in Figure 3 The ITAP has been segmented into 4 section as to identify the areas of the recommended change of the topology, when simulated with the loads associated with walking and bone failure. This has been done as it is imperative that the ITAP bonds to the bone and allows the muscle to congeal around the protruding end. As such the changing topology must be correctly identified and implemented, in different section to reduce the risk of failure of insulation. As the designed in failure mechanic must failure externally of the body. Thus, the segments utilized are Section A, B, C and D theses represent the region of the ITAP: embedded in to the femur that must have a constant surface contact with bone; located at the end of the users stump where the collecting skin, fat and muscle fusses; were the desired failure mechanic is needed to be implemented for a safe failure of the ITAP; were the prosthetic leg is attached to the user, respectively. For the simulation of the design the following assumptions are mad.

The ITAP and femur are both cylindrical, where the femur has an outer diameter of 28 mm and is extruded for 311.5 mm. Whilst there is often a curve in the femur, the tensor faciae latae muscle can reduce the stress acting along the bone and therefore reduce any pronounced stress concentrations acting along the femur [18].At one end of the femur there is a 14 mm diameter hole, centred in the middle, cut to a depth of 160 mm.The material properties of the femur have been assumed to be cortical bone, which has a σ_s_ of 110 and 120 MPa for compressional and tensional forces. The mechanical strength of a femur depends on numerous factors, such as age, porosity and mineral content. However, Marco et al. [19] and Reilly and Burstein [20] found average elastic and shear moduli of human femur specimens which can be used as a preliminary step towards modelling the bone housing ITAP implant. A summary of these mechanical properties is presented by Bird et al. [13].The material used for the ITAP has been set as Ti from the ANSYS’s database, this has a σ_s_ of 930 MPa and an ultimate tensile stress, UTS, of 1070 MPa. The material selection for the computational modelling is in line with the Ti material used for the ITAP, thus the modelling is a comparable study.As the focus of the simulation is on the ITAP, the femur is fixed in place to the pelvis.The femur will not be simulated and the free body diagram (FBD) will only consist of the ITAP and the resulting σ is analysed for the impact on the bone anchor.

The mesh was developed by conducting a mesh sensitivity study. Ten simulations were carried out with different mesh refinements to determine the following σ results:σ_Max_ across the entire geometryσ_Max_ recorded in the core of the ITAP using a set pathway in the centre of the cylinderσ_Max_ recorded in the core of Section B only using a pathway control.

From the study the optimum controls for the simulation where the following;

A face sizing on both ends of each section of the ITAP, measured at 2 mm.A face sizing on the cylindrical surface of each section of the ITAP, measured at 2 mm.An edge sizing on both edges of the ITAP sections, divided by 25.

This process removed errors and many delocalisations within the mesh to ensure each section is uniform and in alignment with each other. However, when reviewing the full geometry and specific section minor desolations where identified. This is due to the simulated model being constructed from separate sections, forming a full geometry, rather than one single section. This purpose of this was to allow an in-depth review of each section of the ITAP and change the failure mechanic within section C of the ITAP.

The simulation boundary locations have been set at the femur joint with the pelvis. Frontal (F_x_) Lateral (F_y_) and Axial forces (F_z_) are acting on the ITAP in the region where the prosthetic would be applied as this is the same area as the patellar surface where the knee would be attached [19], as seen in Figure 4.

The work of Georg et al. identified that the forces acting on the femur are dependent on the location of interest within the bone, the body weight (BW) of the user and finally the position of the motion of walking [21]. As such, the F_x_, F_y_ and F_z_ maximum forces are calculated as 958.9 N, −833.8 N and 3168.6 N, respectively.

### 2.2. Simulated Results of the ITAP Standard Design

The σ_Max_ recorded when the ITAP is subjected to walking loads is shown in Table 1. It can be identified that as the diameter of the safety notch decreases from 14 mm to 5 mm the σ_Max_ increases. It can also be identified that the location of the σ_Max_ within the ITAP changes its position as the size of the notch diameter decreases, moving closer to the notch from the point of insertion. Discrepancies arise when analysing the full geometry’s core and only Section C’s core, this is partly due to minor delocalisation in the mesh within the model, further improvements have been utilised for additional readings. However, it can be identified that the recorded σ in Section C’s core for all simulated models does not exceed the σ_s_ of Ti. Further conclusions can be drawn upon from this preliminary study, these being:The basic ITAP design has regions of σ above 930 MPa, with a σ_max_ of 1175.6 MPa at the A-B joint interface. This is greater than the σ_s_ of the Ti, and could result in a section of the ITAP permanently deforming and making it unsuitable for use.For all simulated geometries the section A σ is greater than both the compressive (110 MPa) and tensile (120 MPa) strength. This only occurs at the very start of the embedded part of the ITAP.

The results indicate that the ITAP design needs further optimisation to reduce the σ at the joint of bone and ITAP to less than the σ_s_ of bone, as well reducing the σ across the whole design so that the σ_Max_ is less than the σ_s_ of Ti. Topology analyses shall be utilised as the ITAP must fail within the core of Section C when the σ generated by external loads are greater than the limits of the system, thereby avoiding irreparable damage to the user. However, the core and ITAP must not fail under standard forces generated in day to day life such as walking, as seen in a 5 mm safety notch were the recorded σ is at 3313.8 MPa.

## 3. Topology Optimisation of for the ITAP Model

In the following section the approach to using topology modifications to the ITAP will be show. Firstly, the ANSYS model will be described followed by the 2D and 3D approaches using MATLAB functions.

### 3.1. ANSYS Model

With a material reduction (50%) the design of the standard ITAP and notched designs have been simulated in ANSYS. In total 20 different simulations where conducted utilising the topology reduction calculations in order to identify key structural information for the IP when underload. The model mass is reduced in areas that are not critical to the mechanical structure, as seen in Figure 5, Figure 6 and Figure 7. The same forces identified in Section 2.2 are used and the fixed supports used are as follows.

The connecting surfaces of the ITAP and bone.The connection of the top surface of the ITAP and bone.The top of Section C joined to Section B of the ITAP (Figure 8B).

### 3.2. 2D Simulations Using MATLAB Functions

In this section a 2D infill for the entirety of the interior of the ITAP geometry is developed. The coding for designing an optimised interior where material can theoretically be removed originated as a 2D MATLAB problem written by Otomori et al. [22]. This has since been utilised to identify if the ANSYS topology studies identify the correct locations to remove non-critical material from the cylindrical cross-section of the ITAP’s external Sections C and D. The volume of material remaining is denoted by V_max_, and is the only variable in these simulations. Different iterations were completed looking at Vmax (50–95% in intervals of 5%), in total 10 different simulations were conducted to complete this section of research. Using MATLAB code for a level set-based topology optimization the assumptions made in the model are as follows;

Function control levelset88 (Nelx, Nely, Vmax, Tau) are usedThe simulation considers the cross-section of the ITAP, external sections away from the bone and fusing muscle.The length X and Y are the length and diameter of the external region of the ITAP, Sections C and D, (represented by Nelx and Nely, respectively).The position of the applied load, interconnectivity and complexity is denoted by Tau and represents the regularization parameters which are set to the recommended value of 2 × 10^−4^ [21].The magnitude of the forces is not considered.The Young’s modulus for the material is set to 1, whereas the Young’s modulus of the voids has been set to 0.Poisson’s Ratio is set at 0.3.

The resulting design structures (Figure 9) show that as the maximum volume of the model increases, the size of the voids forming in the ITAP decrease. This is most prominent in the voids forming at the boundary wall and the voids forming where sections C and D meet. This study has identified where the areas of the topology of the ITAP should be changed depending on the volume of the material used in additive manufactured production processes for complex internal structures. The topology simulation using MATLABs’ applied load only consisted of one shear force, rather than the three forces that are applied to the ITAP in motion. Furthermore, the force cannot be set to the applied loads identified in Section 2.1. As such, the results from this study can only be used as a guideline.

### 3.3. 3D Simulations Using MATLAB Functions

To identify the influence of multiple forces on ITAP topology further simulations were needed. The work of Liu et al. allows for simple 3D models to be designed and simulated, where fixed support and applied loads can be controlled with multiple forces being utilised at a single location [23]. Using cubic models three separate simulation models have been studied looking at Sections C and D of the ITAP. These consist of a 85 mm by 14 mm by 14 mm model (Standard), as well as two models with cross-sections through the centre of the model at 85 mm by 14 mm by 1 mm (Vertical) and 85 mm by 1 mm by 14 mm (Horizontal). A 3D multsicale topology optimization code for lattice microscale response is used (top3d, Nelx, Nely, Nelz, Volfrac, Penal, Rmin) and the assumptions made for this study are as follows:The dimensions of the geometry are individual to each simulated model; this consists of Nelx, Nely and Nelz.Volfrac represents the volume fraction limit of the simulations; the range used for this study will be concordant with the 2D MATLAB study ranging between 50% and 95% in intervals of 5%.Penal represents penalization procedure and is usually referred to as the Solid Isotropic Material with Penalization (SIMP), which is used as a binary solver where penal > 1. In this study it is being set as a constant of 1 following the recommendations of the author.Rmin is the filter size which is the distance between the centroid of element i and element j of the model, this has been set to a value of 1, so there is a millimetre between the centre of each element.The Young’s modulus of the material has been set to 1 while the Young’s modulus of the voids has been set to the value of 0.Poisson’s ratio is set to 0.3.The fixed supports are distributed evenly over the following coordinate positions fixed in all three axes:
○Standard: X1 = 0, X2 = 0, Y1 = 0, Y2 = 14, Z1 = 0, Z2 = 14.○Vertical: X1 = 0, X2 = 0, Y1 = 0, Y2 = 14, Z1 = 0, Z2 = 1.○Horizontal: X1 = 0, X2 = 0, Y1 = 0, Y2 = 1, Z1 = 0, Z2 = 14.The applied forces are Axial 3168.63 N, Lateral −833.85 N and Frontal 958.93 N in the following axes Fx, Fy, Fz, respectively.The following coordinates and forces have been used in the load distribution:
○Standard: Fx, Fy and Fz at X1 = 85, X2 = 85, Y1 = 0, Y2 = 14, Z1 = 0, Z2 = 14.○Vertical: Fx and Fy at X1 = 85, X2 = 85, Y1 = 0, Y2 = 14, Z1 = 0, Z2 = 1.○Horizontal: Fx and Fz at X1 = 85, X2 = 85, Y1 = 0, Y2 = 1, Z1 = 0, Z2 = 14.

The visual results of the simulations can be seen in Figure 10 for selected volume percentages of material.

The MATLAB 3D simulations provide the most detailed results. From the study, the areas where material can be removed to save weight due to the minimal σ present is consistently located at two positions. These being at the centroid of the fixed support at the boundary wall where the ITAP’s Sections B and C meet. The next area where material can be removed is located within Section D, 57 mm < x < 85 mm see Figure 10A, where material is being removed on the opposite side of the applied Lateral and Frontal loads. These two positions of material reduction are concordant with both the ANSYS and 2D MATLAB simulations as well as all three models of the 3D MATLAB simulations. The size of the voids forming is dependent on the volume percentage of material being simulated. The results from the Standard models identify how the ITAP could be shaped to meet the individual’s body characteristics using the loads calculated. The results identify the areas of material that are less important to the structural integrity of the external sections of the ITAP. The Vertical and Horizontal simulations look at the cross-section of the ITAP with the Axial and either Lateral or Frontal forces applied, Furthermore, the remaining internal structure mimics the internal structure found in bone, Figure 10, forming pathways in-between voids in the ITAP, similar to the findings of both Otomori and Wu with their respective studies and coding [22,24]. Both the Vertical and Horizontal results show this pattern. The voids are geometrically triangular in design as they are extremely capable of distributing σ within their shape [24]. There is minimal variation between the two sets of results, and this is expected as the forces being applied are similar in size. As the percentage of the material used increases, the definition of the voids and interconnecting pathways decreases. For these study 30 simulations were conducted utilising 10 different volume fraction limits for three different tested geometries.

## 4. Topology of the ITAP Design under Extreme Loads

In this study the shear force being applied is set to the interconnecting surface between the prosthetic and the ITAP, represented by F_y_ as −4000 N. There is a secondary axial force of 416.9 N applied, denoted by F_z_. This has been derived from half of the user’s weight resting on the ITAP, from the 85 kg test individual. From analysing the results of the simulation, the location of the σ_Max_ can be seen to be at the joint between Section A and B, where the ITAP leaves contact with the bone anchor, as seen in Figure 8B. This is the same location as in the walking simulations. The only difference is the position on the circumference of the joint of Sections A and B, as the forces are acting in different directions. The σ_Max_ has been recorded above the σ_s_ of Ti and both the compressive and tensile σ_s_ of bone; therefore, a fracture has a high chance of occurring under this load. From these findings, it is recommended that further study is needed into the design of the ITAP to remove the risk of critical failure of the bone when exposed to extreme loads.

### 4.1. ANSYS Topology of the ITAP with Extreme Loads Applied

For this study the assumptions and controls are the same as in Section 3.1. The results identify that voids are forming in the same locations as in the walking simulations, these being at the boundary walls between sections and fixed supports. Material is also being removed along the sides of the ITAP in the same manner as the walking simulations. However, the main differences in the results is the voids forming within Section D, where more material remains in these locations to disperse the σ throughout the model from the applied loads. As such 4-separate standard ITAP models where simulated utilising ANSYS’s high computational topology modelling process. As seen in Figure 11 and Figure 12.

### 4.2. 3D MATLAB Analysis of the ITAP with Extreme Loads Applied

For this study the assumptions and controls are the same as in Section 3.3 here the geometries used are the Standard and Vertical models. The simulations have been run between 50 and 95% of material in intervals of 5%, resulting in 30 separate geometries being developed for simulations. This study has identified the critical areas of material within the ITAP’s Sections C and D at different volume percentages of material, by removing material that had the lowest σ concentration within the geometry. Common trends have been identified in these simulations. This has included: voids forming at the centre of the fixed support; divots manifesting within Section D; triangular interconnecting pathways forming within the voids in the centre of the models; and critical material kept along the boundary walls, most prominently the wall with the force directed towards it. All these observations are identical to the walking simulations 3D MATLAB study. As the percentage of the volume increases, the voids decrease in size as expected, but the pattern built into the structure is constant with the exception of the vertical model at 95%. The results from the latter simulation is sporadic because the material removed is not concordant with previous iterations.

## 5. Discussion

### 5.1. Similarities of the ANSYS, 2D and 3D MATLAB Topology Studies on the ITAP for Walking Simulations

Each simulation process has identified which regions of material are critical for the distribution of σ within the ITAP and what material can be removed to reduce the mass of the ITAP. As stated before, these results identify the material that can be removed as it is not critical to the structure of the ITAP. As such, the inverse of the results should be studied to correctly design the failure features of the ITAP, so that the device can be correctly used in day-to-day life, but will fail in a designed manner when exposed to extreme loads. Looking at Figure 13 which shows the visual results for each simulation method for sections C and D of the ITAP at 50% by volume of material, a clear pattern can be identified for further study. There are similarities in the reduction of material in each process, most notably the reduction of a large proportion of material at the fixed support of Section C, which is present in all models. Other significant similarities are the reduction of material identified in Section D, where material is removed in the opposite direction of the Frontal and Lateral forces for the divot. This is most prominently observed in both the ANSYS and 3D MATLAB simulations as the magnitude of the forces has been directly controlled and adjusted accordingly. The inverse of the results shows that the σ found within these locations is not substantial and therefore not critical for material reduction. The main areas where reduction could be focused on to achieve a controlled failure are the two quarters of the models that do not have any change in design; this can most prominently be seen in D in Figure 13 as well as A and B. Both the ANSYS and the Standard 3D MATLAB simulations are very similar in the locations of material reduction; the main differences arise due the starting geometry used.

The 2D MATLAB simulated model C is the most diverse, because the algorithm calculating the theoretical structure that could be capable of withstanding shear forces only at a point location rather than a dispersed load that is controlled. As such, the 2D MATLAB simulations do not work out the best structure to an individual load parameter, but rather work out the optimum geometry for shear loads being applied.

The results from this study have identified which internal structures should be utilised to distribute σ while saving weight. There are similarities in the simulated models such as the aforementioned void forming at the top of Section C in the ITAP, as well as the voids in the main body of the section being triangular based; this is similar to the results of E and F in Figure 13. However, from analysing and inverting the results provided from the use of Otomori’s code, the areas of critical importance for distributing σ when designing the internal topology for shear σ are the boundary walls of the model. This is concordant with the range of percentages of volume simulated in this study. The resulting design structures (Figure 9) show that as the maximum volume of the model increases, the size of the voids forming in the ITAP decrease. This is most prominent in the voids forming at the boundary wall and the voids forming where sections C and D meet. This study has identified where the areas of the topology of the ITAP should be changed depending on the volume of the material used in additive manufactured production processes for complex internal structures. The topology simulation using MATLABs’ applied load only consisted of one shear force, rather than the three forces that are applied to the ITAP in motion. Furthermore, the force cannot be set to the applied loads identified in Section 2.1. As such, the results from this study can only be used as a guide. Bye analysing the studies it can be seen that only critical material is left, thus it can be inferred that this is the area where design changes are required for a controlled failure. This is concordant with the work of Bird et al. [13], where the location and depth of the safety notches affected the ITAP performances.

The findings from the 3D Vertical and Horizontal MATLAB studies support this claim. However, the findings identify that one section of the material has more importance than the other. In Figure 13 both E and F’s cross-section of the ITAP has a thicker boundary wall on one side, located in the direction of F_y_ or F_z_, respectively. Both these locations are more regular and thicker than the opposite side of the ITAP. Inferring that the thicker side is more important for structural integrity and therefore should be the focus location when designing in the controlled location of failure. In total 60 different simulations were conducted to identify and consolidate the critical material needed for the structural stability of the ITAP when simulated with walking loads.

From the results of these studies (Table 2), the following conclusions and recommendations can be made for further development in simulation and physical testing of the topology of the ITAP:Integrate a stress raiser on the ITAP at the end of Section B at the bone anchor, with appropriate fillets on either side to increase contact with bone and reduce residual σ under load at the connection between the ITAP and anchor.The external walls around the ITAP are key to the stability of the design; minimal removal in any direction will weaken the model. However, removing critical material from the combined direction of the Lateral and Frontal forces will introduce a significant weakness in design. Removal of material in the opposite direction will also have a significant reduction in strength.The internal topology of the ITAP should be designed following the results of the study’s triangular-based patterns for stability, with AM production processes in mind.Any design models should be simulated using the forces utilised in these studies as well as simulation with forces that would fracture the femur.

### 5.2. Discussion and Similarities of the ANSYS and 3D MATLAB Topology Studies on the ITAP when Exposed to Extreme Loads

From analysing both the data from ANSYS and 3D MATLAB topology studies for an ITAP exposed to extreme loads, see Figure 14, that are expected to cause fracture to the femur, the following observations and conclusions can be drawn:The σ within the ITAP is significantly greater than the capacity of the anchor. Therefore, design iterations are needed to remove the failure of bone, and the location of the σ_Max_ is the same as the walking simulations in the anchor.The external walls, perpendicular to the forces are critical to the structural integrity of the ITAP. This is more prominent in the wall to which the shear force is directed.Voids in the centre are not critical to the structure but there is a triangular interconnecting pattern similar to the structure of bone which is required for structural stability within the models [21,23,24].

The findings from these studies are identical with the walking simulations, indicating that critical material should be removed from the model on the boundary wall, to which the shear the force is directed. However, the difference in the positioning of the forces is subject to change for the extreme loads because impact could occur from any direction. Thus, a failure feature must be designed that is capable of failing from any direction of shear force. This was concordant with all 34 simulations. The similarities of the location and σ_Max_ recorded for both the walking and extrema loads can be seen in Table 2. This also identifies the mass reduction calculated from the ANSYS calculations.

## 6. Conclusions and Recommendations for Future Work

The ITAP is an alternative prosthetic product that has the potential to remove the inflammation and pain from traditional prosthetics while replicating the feeling of having a leg via a direct skeletal implant, thus improving the quality of life of the user. The inherent risk of using an ITAP is that because it is attached to the bone any σ on it receives is transferred to the bone. To prevent further limb damage a failsafe safety notch design is considered. The optimisation challenge is to ensure the ITAP is functional for normal use while allowing for a designed failure if excessive forces are experienced. The following conclusions are made.

Due to advances in medical practises, CT scanning has been performed identifying the condition and strength of the bone anchor. Thus, allowing for clear understanding of how an ITAP will integrate within the user after an amputated femur. The acting loads applied to the ITAP when in use, have been ascertained, allowing for design and manufacture using AM methods to incorporating topology modifications bespoke to the users’ lifestyle.The topology studies of the ITAP have been performed using ANSYS and 2D and 3D MATLAB models. The use of the 2D modelling has quick results that allows for rapid modelling, with alternative length of prosthetics. However, there is no control on the magnitude of the applied load. As such it is only recommend for quick analysis for an idea of where material is critical with in a cross section. The use of the 3D code is better suited to live modelling as there is more control on both the material properties and applied loads. The benefit of this is a visual guide to the critical material that is accurate to the model. However, the problem is the simulation does not yield numerical results and have limited geometry modelling. As such it is best suited for quick accurate visual modelling of cross-sections. In total 10 2D and 60 3D Matlab simulations where carried out within this research, all results yield from these simulations gave clear insight into the topology of the ITAP under load.ANSYS code has been developed for advances topology modelling. This has allowed for in-depth studies into the ITAPs geometry looking at the core material and σ that are being applied to the ITAP. This has allowed for in-depth review of the ITAPs critical material and local maximum stresses when modelling the applied loads. The downside of this simulation is that it is computationally expensive and time consuming, a large proportion of time was utilised simulation the 24 different ANSYS topology models.The topology studies for both the walking and extreme loads have identified critical areas of material within the ITAP that are needed for σ to dissipate through the product. Through the use of ANSYS, 2D and 3D MATLAB models, it can be concluded that when the ITAP is exposed to walking loads, the critical material is solely located in line with the summation of the walking loads, which is set in a fixed direction. It is also critical that the section of the ITAP that meets the bone anchor does not exceed the bone σ_s_ of 110 MPa as this will cause further injury to the user.The critical support material identified from the topology studies is directly in line with the extreme loads. However, in use as an extreme load can be applied from any direction, and all of the ITAP geometry can be assumed as critical so a controlled failure safety notch feature is adopted. As such it can be stated that a developed ITAP must not exceed the σ_s_ of bone but also be capable of withstanding loads for daily use like walking. For the failsafe to work the σ within the core section C must exceed the material UTS when exposed to extreme loads in any direction.

It is recommended that further research is undertaken on the design of the ITAP’s failure feature utilising the topology study in this paper so that a safety design can be successfully developed for each individual case. Physical testing is also highly recommended as to ascertain the strength compression on different volume fraction of the ITAP, when tested with walking loads and destructive forces. Further simulation on different loads including running and jumping to fully facilitate the user’s mobility would gain an insight in the loading capabilities of the ITAP.

## Figures and Tables

**Figure 1 micromachines-12-00298-f001:**
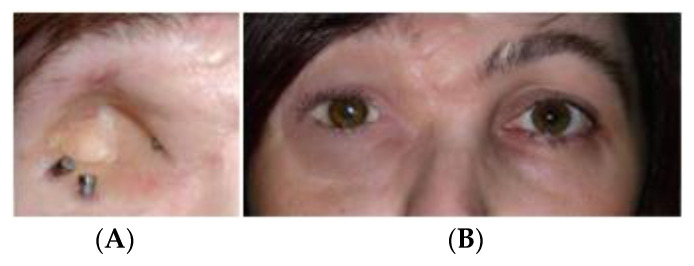
(**A**) ITAPs embedded into the skull for a prosthetic eye, (**B**) Prosthetic eye mounted on the ITAPs into the skull [11].

**Figure 2 micromachines-12-00298-f002:**
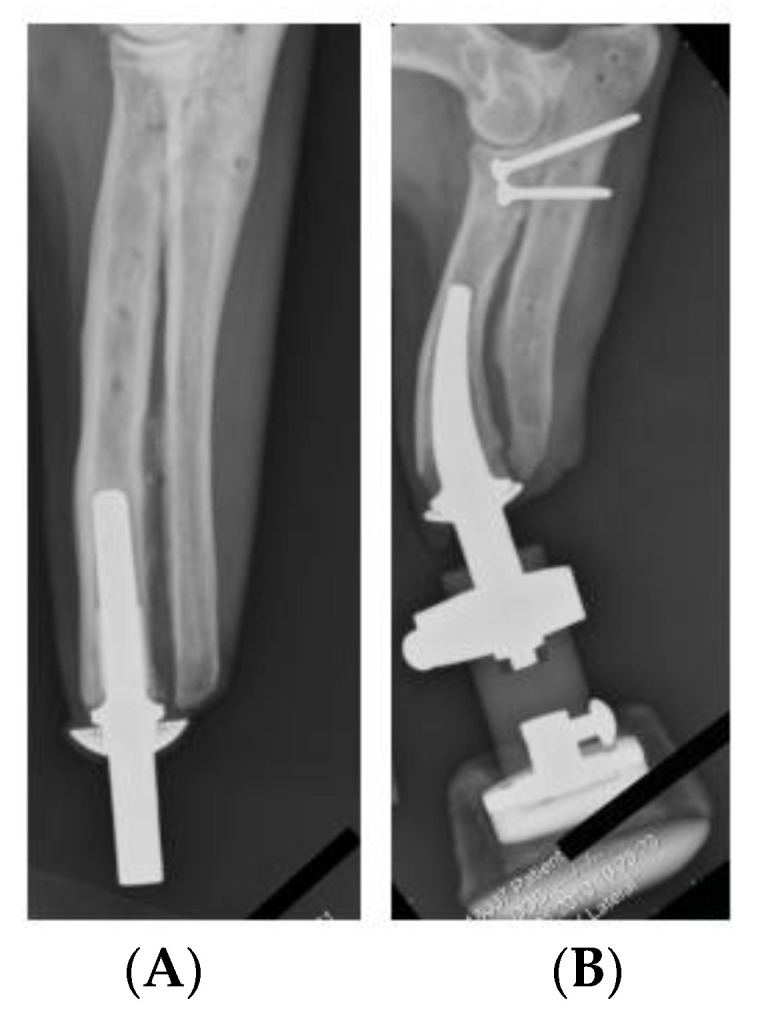
(**A**) Canine prosthetic ITAP embedded into its front left leg. (**B**) A Canine bespoke ITAP embedded into its front right leg [15].

**Figure 3 micromachines-12-00298-f003:**
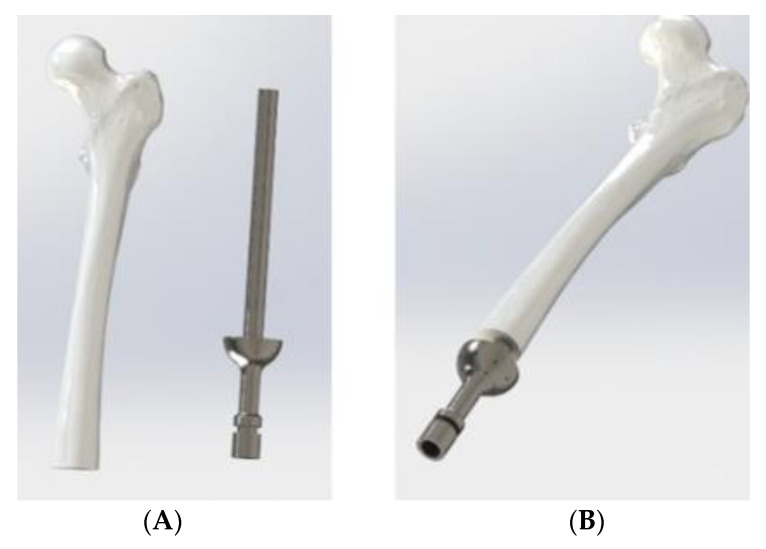
(**A**) Quarter Amputated Femur with accompanying ITAP CAD models, (**B**) Femur and ITAP assembly [17].

**Figure 4 micromachines-12-00298-f004:**
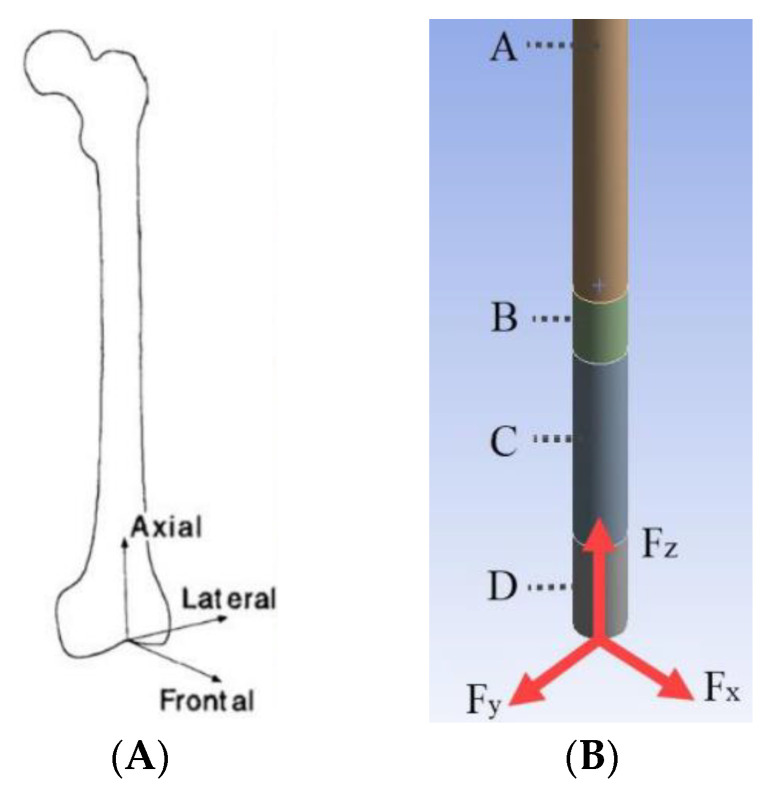
(**A**) Forces acting upon the femur along the bone’s structure [21], (**B**) FBD of the ITAP, built into Sections A, B, C and D.

**Figure 5 micromachines-12-00298-f005:**
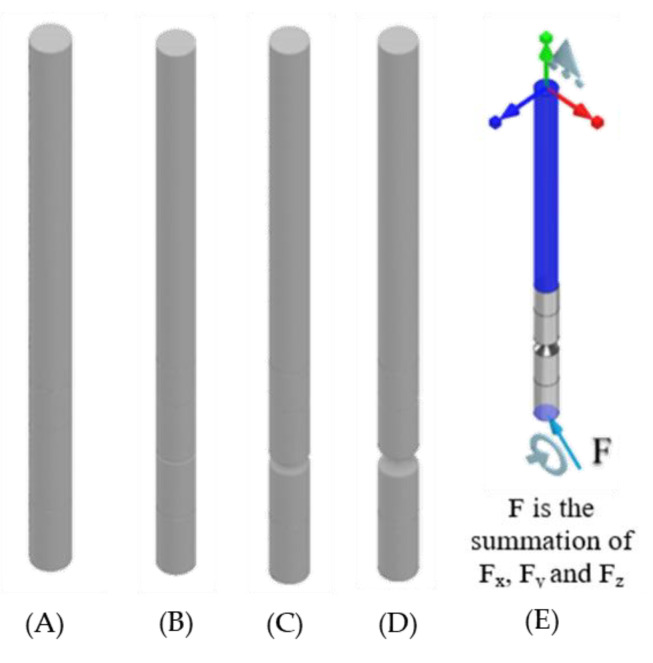
Fixed contact between the bone-anchor and ITAP, (**A**–**D**) Standard ITAP to 5 mm safety notch topology reduction models results, (**E**) FBD of the applied walking loads.

**Figure 6 micromachines-12-00298-f006:**
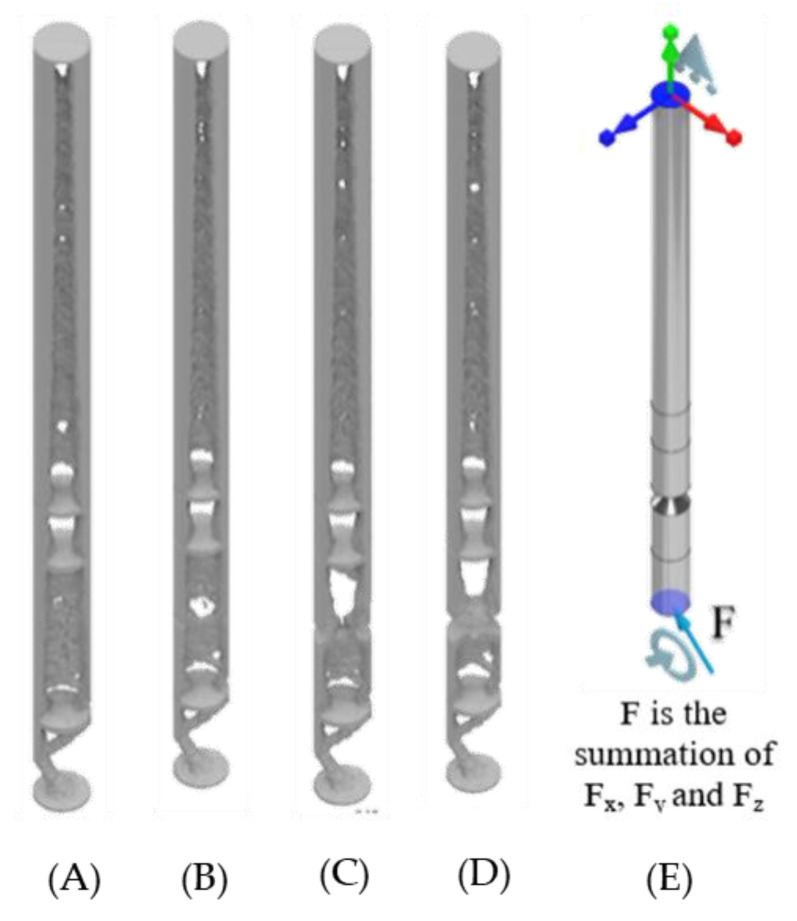
Fixed contact for the head of the bone-anchor to ITAP, (**A**–**D**) Standard ITAP to 5 mm safety notch topology reduction models results, (**E**) FBD of the applied walking loads.

**Figure 7 micromachines-12-00298-f007:**
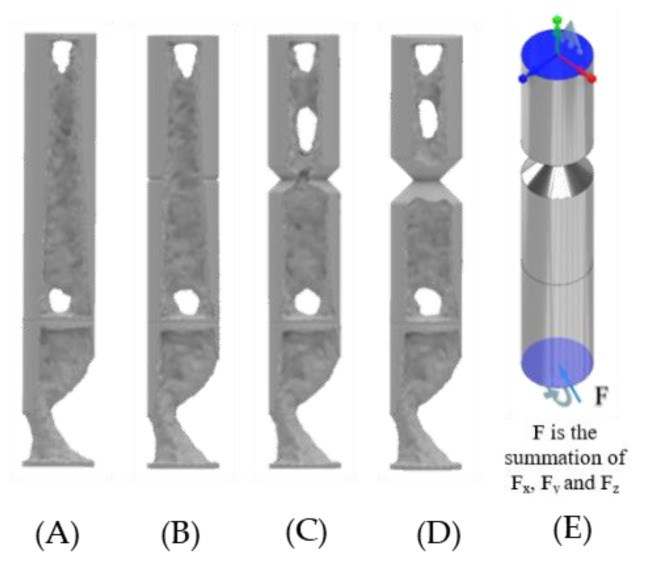
Fixed contact for the head of Sections B and C of the ITAP, (**A**–**D**) Standard ITAP to 5 mm safety notch topology reduction models results, (**E**) FBD of the applied walking loads.

**Figure 8 micromachines-12-00298-f008:**
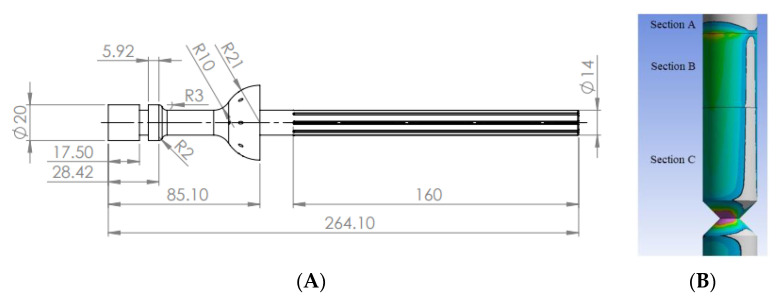
(**A**) Initial ITAP designed, (**B**) σ distribution in the 5 mm safety notched ITAP.

**Figure 9 micromachines-12-00298-f009:**
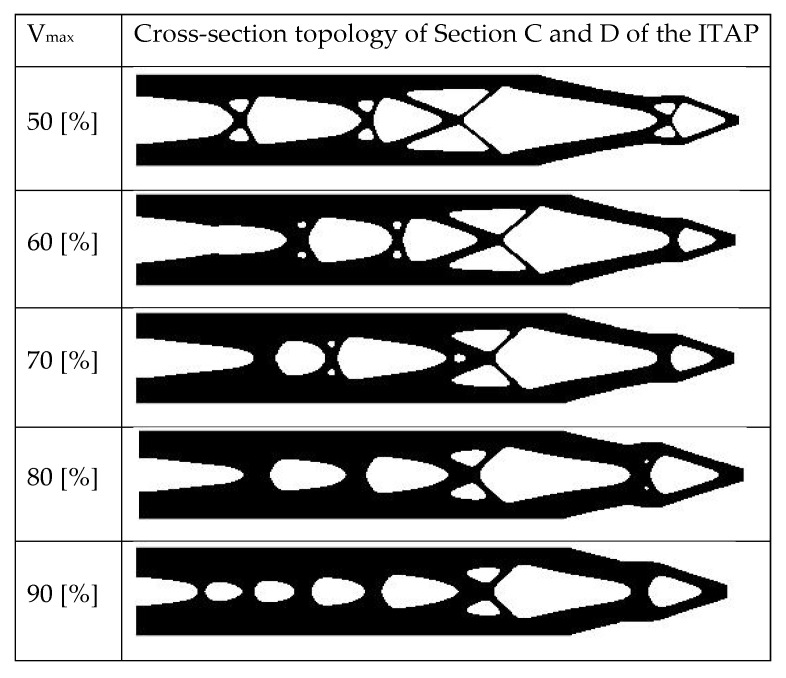
Simulated results of the 2D MATLAB topology simulations identifying the theoretical areas of material reduction of 50 to 95%.

**Figure 10 micromachines-12-00298-f010:**
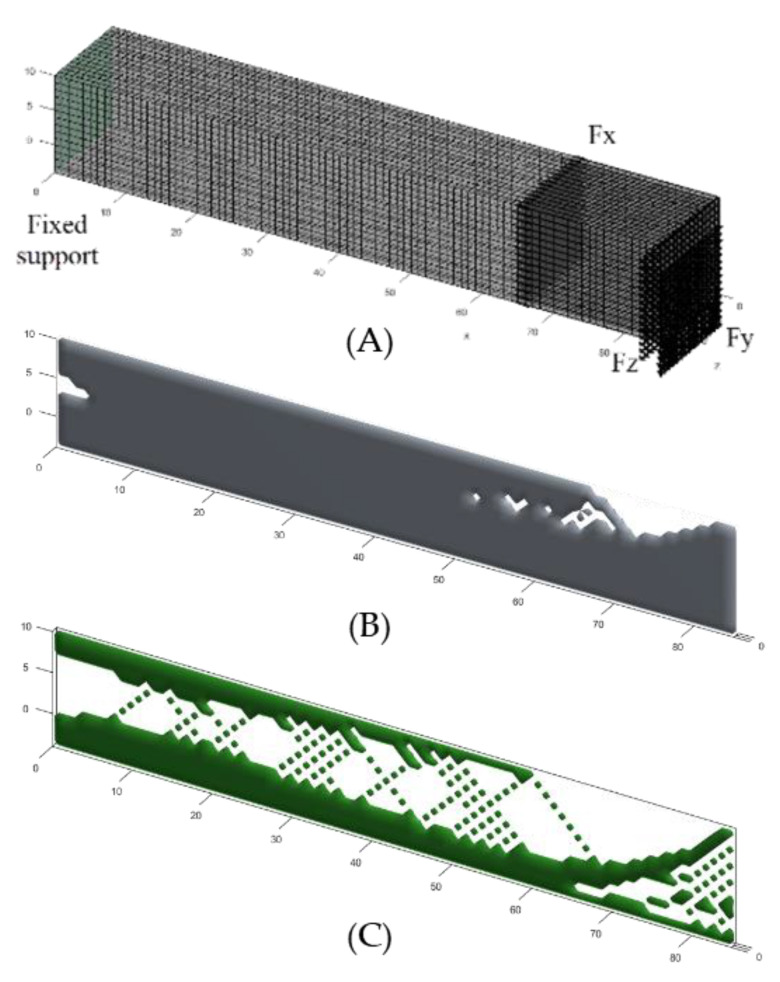
(**A**) FBD of the 3D MATLAB topology simulation utilising the external Sections, C and D, of the ITAP, dimensions of 85 mm by 14 mm by 14 mm with applied loads, (**B**) Vertical model using F_x_ and F_y_ 95% volume of material, (**C**) Vertical model using F_x_ and F_y_ 55% volume of material.

**Figure 11 micromachines-12-00298-f011:**
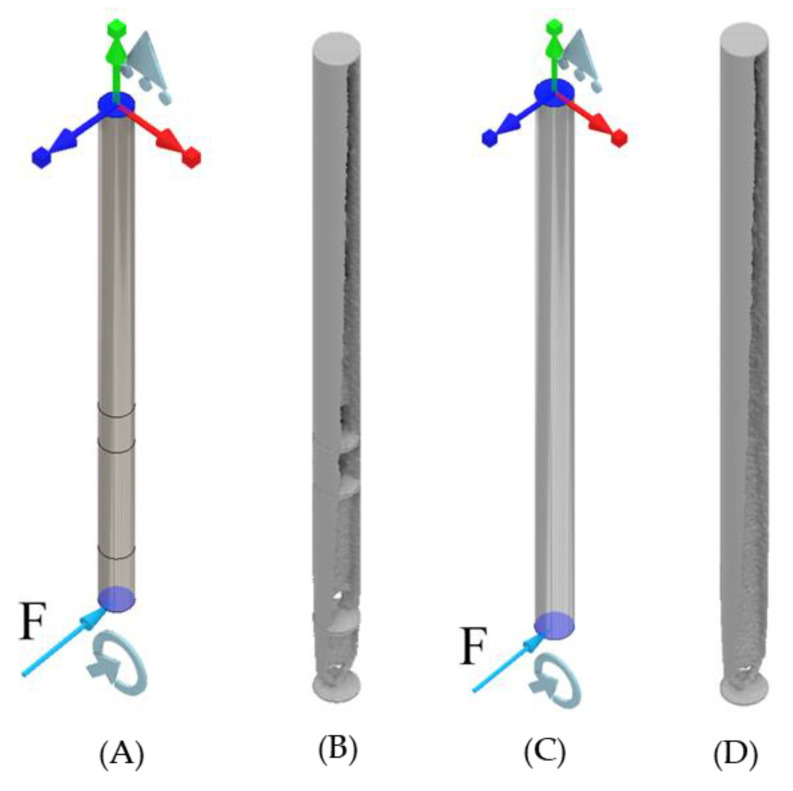
Topology simulations of the full standard ITAP with extreme load F, which is the amalgamation of F_y_ and F_z_ at 50% volume of material, (**A**) FBD of the ITAP in Sections, (**B**) The resulting topology of the individual Sections, (**C**) FBD of the ITAP constructed in one Section, (**D**) The resulting topology of the full ITAP.

**Figure 12 micromachines-12-00298-f012:**
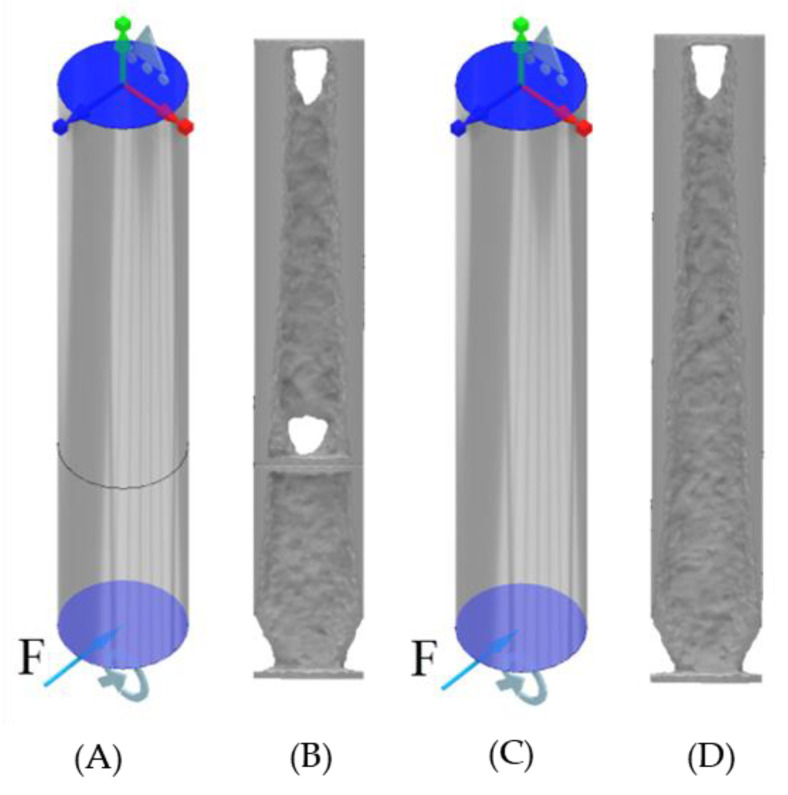
Topology simulations of Sections C and D of the standard ITAP with extreme load F, which is the amalgamation of F_y_ and F_z_ at 50% volume of material, (**A**) FBD of the ITAP in Sections, (**B**) The resulting topology of the individual Sections, (**C**) FBD of the ITAP constructed in one Section, (**D**) The resulting topology of Sections C and D.

**Figure 13 micromachines-12-00298-f013:**
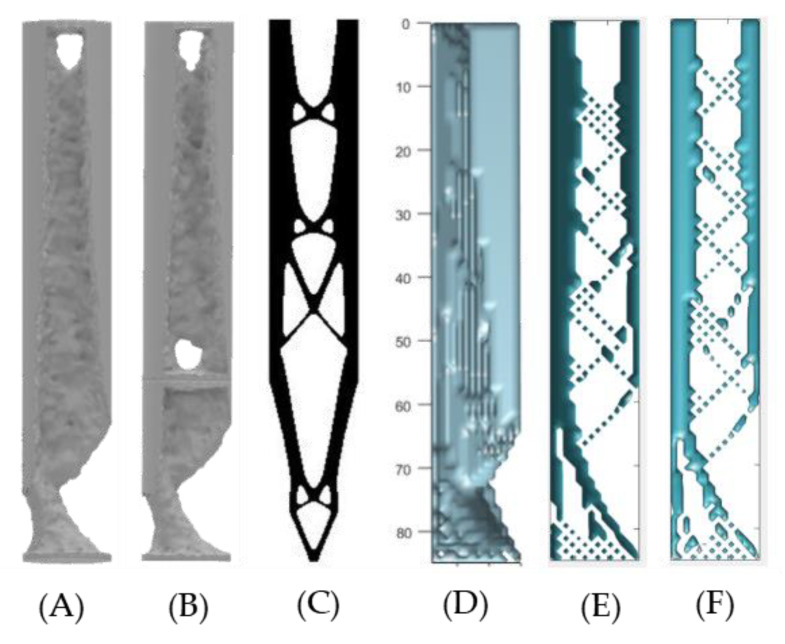
Visual comparison between topology simulation of Section C and D of the ITAP at 50% volume of material, (**A**) one solid model using ANSYS’s control, (**B**) 2 sections joined together using ANSYS’s simulation controls, (**C**) 2D MATLAB simulations with only the shear force being applied to identify the potential areas of material reduction, (**D**) 3D MATLAB standard model, (**E**) 3D MATLAB vertical model, (**F**) 3D MATLAB horizontal model.

**Figure 14 micromachines-12-00298-f014:**
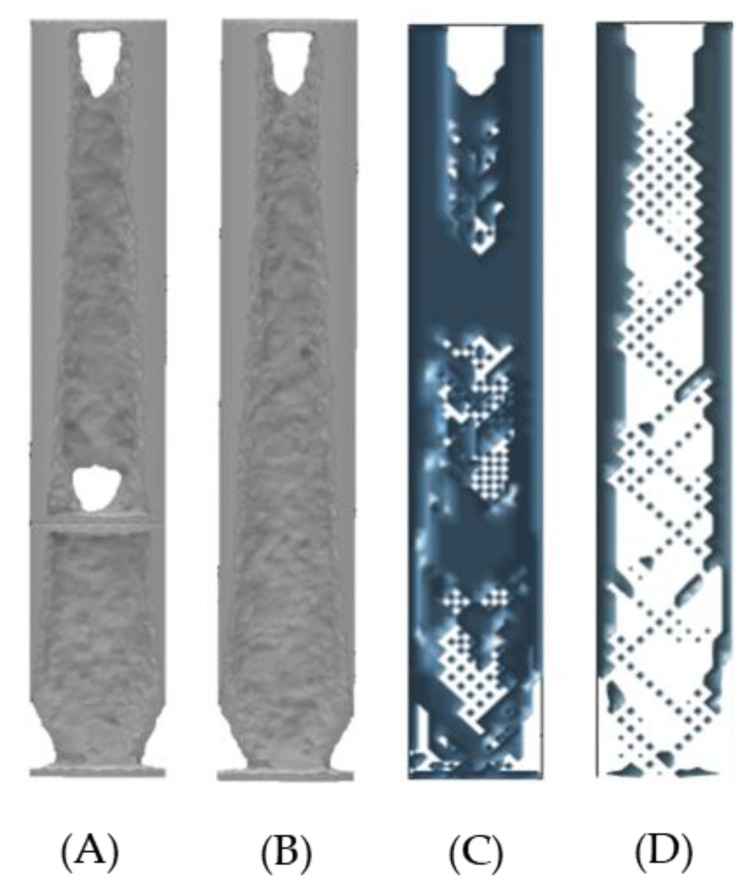
Visual comparison between topology simulation of Sections C and D of the ITAP at 50% volume of material, (**A**) one solid model using ANSYS’s control, (**B**) 2 Sections joined together using ANSYS’s simulation controls, (**C**) 3D MATLAB standard model, (**D**) 3D MATLAB vertical model.

**Table 1 micromachines-12-00298-t001:** Simulated σ_Max_ of the ITAP designs.

ITAP Designs	Standard	13 mm Safety Notch	9 mm Safety Notch	5 mm Safety Notch
σ_Max_ [MPa] In the model	1175.6	939.5	939.5	3313.8
σ_Max_ [MPa] Core of the ITAP	79.5	74.5	146.1	1003.6
Vertical height from applied loads, z domain [mm]	104.5	104.5	55.1	55.0
σ_Max_ [MPa] in Section A	570.3	735.6	735.6	735.6
σ_Max_ [MPa] Core of Section A	67.1	72.1	72.1	72.1
Vertical height from applied loads, z domain [mm]	107.4	107.4	107.4	107.4
σ_Max_ [MPa] in Section C	422.6	474.5	861.6	3313.8
σ_Max_ [MPa] Core of Section C	29.5	30.8	150.3	895.4
Vertical height from applied loads, z domain [mm]	64.7	49.9	53.3	59.0

**Table 2 micromachines-12-00298-t002:** Simulated σ_Max_ and mass reduction of the ITAP when studied at 50% topology reduction.

σ_Max_ and Mass Reduction	Walking Loads				Extrema Load
ITAP Designs	Standard	13 mm Safety Notch	9 mm Safety Notch	5 mm Safety Notch	Standard
σ_Max_ [MPa] In the model	1175.6	939.5	939.5	3313.8	3220.9
σ_Max_ [MPa] Core of the ITAP	79.5	74.5	146.1	1003.6	211.7
Vertical height from applied loads, z domain [mm]	104.5	104.5	55.1	55.0	104.5
σ_Max_ [MPa] in Section A	570.3	735.6	735.6	735.6	1813.7
σ_Max_ [MPa] Core of Section A	67.1	72.1	72.1	72.1	215.0
Vertical height from applied loads, z domain [mm]	107.4	107.4	107.4	107.4	107.4
σ_Max_ [MPa] in Section C	422.6	474.5	861.6	3313.8	1264.0
σ_Max_ [MPa] Core of Section C	29.5	30.8	150.3	895.4	64.8
Vertical height from applied loads, z domain [mm]	64.7	49.9	53.3	59.0	69.2
Mass of full ITAP [g]	187.8	187.7	186.7	184.5	187.8
Mass of full ITAP after 50% topology reduction [g]	93.91	93.8	93.3	92.2	93.9
Mass of Section C & D of ITAP [g]	60.5	60.4	59.4	57.2	60.5
Mass of Section C & D of ITAP after 50% topology reduction [g]	30.2	30.2	29.7	28.6	30.2

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
