# Peer review of "The Micro Topology and Statistical Analysis of the Forces of Walking and Failure of an ITAP in a Femur"

_micromachines, 2021, doi:10.3390/mi12030298_

Round 1

Reviewer 1 Report

The paper is interesting and forms a consistent whole. However, I suggest that the Authors take the following comments into account:
 - to what extent does a material selected from the ANSYS software database correspond to the material used to make an ITAP?

 - in lines 181-186 the Authors describe the selection of the optimum FEM grid, but in line 198 they indicate the grid delocalization as one of the simulation results errors. Please describe the FEM grid adaptation in more detail.
- how much does the ITAP geometry change affect the reduction of weight?

Author Response

Comment 1 - to what extent does a material selected from the ANSYS software database correspond to the material used to make an ITAP?

Response. We have added additional text to page 5 (highlighted) to add more clarity to this point.

Comment 2 - in lines 181-186 the Authors describe the selection of the optimum FEM grid, but in line 198 they indicate the grid delocalization as one of the simulation results errors. Please describe the FEM grid adaptation in more detail.
Response. We added a new section on page 4 that describes the FEM adaptation

Comment 3- how much does the ITAP geometry change affect the reduction of weight?

Response. We have added table and this provides detail of the weight saved.

Reviewer 2 Report

Need to add some graphs of results after the section with the simulations. At least the table of results should be moved after the simulations. Need structure of intro methods results discussion conclusions

Please address notes embedded in attached file. 

Author Response

Comment 1. Need to add some graphs of results after the section with the simulations. At least the table of results should be moved after the simulations.

Response. We have now added a table of results on page 16.

Comment 2. Need structure of intro methods

Response. We have now added a new title (Experimental Methodology) this section covers the base line set of planning and experimental data prior to the topology experiments.

Comment 3. Need structure of results discussion conclusions

Response. We have re arranged the paper. The new section 3. Covers the Topology optimisation of for the ITAP model for the 3 topology methods. This is then followed by section 4. Topology of the ITAP design under extreme loads. We decided to separate 3 and 4 as this would be cumbersome to the reader. We have also added a discussion section which covers 2 parts. The first discusses Similarities of the ANSYS, 2D and 3D MATLAB topology studies on the ITAP for walking simulations. This is then followed by the observed differences for extreme loads. The changes to the structure were necessary as observed by the reviewer, and now the paper has more clarity for the reader.

Reviewer 3 Report

Paper deals with important and prospective implementation of ITAP, however, limiting the analysis only on stress induced by static forces alone, may be prove misleading. Shear force is certainly one of the critical loads, however, dynamic and cyclic torques induced throughout the walking were neglected in the paper. Justification should be provided.

Furthermore, stability of the ITAP in the bone structure depends on the bonds established in osseointegration process, as well as on the forces/torques that are determined to act. However, orientation of coordinate axes were not defined appropriately, it should be improved so the reader may identify the meaning of x, y and z indexes. Is the coordinate system fixed, or if not, how to interpret such orientation? What are the conditions and in which locomotor gait stance were the forces determined? 

Use of symbols of stress σ throughout the paper without proper differentiation of their particular meaning may be confusing, since the readers can be other than mechanical engineers that are familiar with it (σ, σs, σmax). At least short description of their meaning and estimated values should be included. If comprehended that the bone properties are not consistent, are dependant to mineral composition and may be different for other subjects, how consistent and repeatable are the findings?

Femur generally is not cylindrical, but as assumption can be approved, but its longitudinal axis is not straight, especially if longer section is considered. The consequences of femur normal curve should be justified in the paper.

Differences in properties of bone and Ti can be another cause of failure of ITAP or any other implant/prosthesis, therefore, authors are advised to address this issue in the assumptions. Many medical complications may be caused by this issue, which is not obtainable by numerical simulation.

Throughout the paper, numeration of headings/subheadings are not consistent and consequent, it should be checked and corrected.

Another issue is in the headings is that sometime is just D, instead of 2D or 3D, respectively, that authors should correct and harmonize with the content below it.

Forces and other values should be justified and clarified - why it is necessary to present force value as: 416.925 N, with 3 decimal places? Same goes for many other physical values throughout, especially for even larger values?!?

"...damage the user..."? - not traumatize, hurt or injure?

References are partially correctly listed, therefore, the references: 5, 7, 9-12, 20, 21 should be corrected, the journal/source name is replaced by Elsevier, or even not mentioned at all.

There is no references from the journal MM, although the references are covering recently published papers. It is recommended but not required to cite papers from the journal (this confirms paper compatibility with journal aims & scope, and readers interest in published paper).

Dynamical rather than static analysis is recommended for future consideration of ITAP functionality.

Numerical simulations and analyses are great tools, but the true problem of osseointegration and effective prosthetic interventions requires comprehensive understanding of medical devices design and their implementation in living tissue.

Author Response

Rev 3

Comment 1. Paper deals with important and prospective implementation of ITAP, however, limiting the analysis only on stress induced by static forces alone, may be prove misleading. Shear force is certainly one of the critical loads, however, dynamic and cyclic torques induced throughout the walking were neglected in the paper. Justification should be provided.

Response. In this simulation, the acting forces were calculated at the most intensive situation where all weight of the user is position on the ITAP. Following the work of Georg et al, it has been identified that in this loading situation “The moments are assumed to be negligible because the influence of BW was constantly recorded at zero.” We hope this clarifies this.

Comment 2. Furthermore, stability of the ITAP in the bone structure depends on the bonds established in osseointegration process, as well as on the forces/torques that are determined to act. However, orientation of coordinate axes were not defined appropriately, it should be improved so the reader may identify the meaning of x, y and z indexes. Is the coordinate system fixed, or if not, how to interpret such orientation? What are the conditions and in which locomotor gait stance were the forces determined?

Response. We have added an image (fig 4) and additional text that shows the relationship of Fz Fy and Fx to the ITAP. This clearly establishes what each force is doing and what plane it is moving in.

Comment 3. Use of symbols of stress σ throughout the paper without proper differentiation of their particular meaning may be confusing, since the readers can be other than mechanical engineers that are familiar with it (σ, σs, σmax). At least short description of their meaning and estimated values should be included. If comprehended that the bone properties are not consistent, are dependent to mineral composition and may be different for other subjects, how consistent and repeatable are the findings?

Response. Hopefully the additions to the paper provide more clarity on the stresses. For the point about bone properties, we have mentioned in the text issues that relate to bone such as age. The point is a good one and further research has been carried out that address some of the wider challenges of the ITAP. However, due to space limitations we have not included them is this paper.

Comment 4. Femur generally is not cylindrical, but as assumption can be approved, but its longitudinal axis is not straight, especially if longer section is considered. The consequences of femur normal curve should be justified in the paper.

Response. This is true, but as the supporting muscle provide counter torque and forces the structure can be largely assumed to be cylindrical as any stress concentrations caused by the curve of the femur. This is clarified with a reference in lines 177-179.

Comment 5. Differences in properties of bone and Ti can be another cause of failure of ITAP or any other implant/prosthesis, therefore, authors are advised to address this issue in the assumptions. Many medical complications may be caused by this issue, which is not obtainable by numerical simulation.

Response. Variations in the properties are highly likely, however average properties found by prior research were used as an initial step for the sake of this research. In future work, variation in bone and Ti properties will need to be analysed. This is clarified if lines 183 – 188 and 190-192.

Comment 6. Throughout the paper, numeration of headings/subheadings are not consistent and consequent, it should be checked and corrected.

Response. We have modified the structure of the paper with an improved layout and better aligned headings

Comment 7. Another issue is in the headings is that sometime is just D, instead of 2D or 3D, respectively, that authors should correct and harmonize with the content below it.

Response. This is now resolved

Comment 8. Forces and other values should be justified and clarified - why it is necessary to present force value as: 416.925 N, with 3 decimal places? Same goes for many other physical values throughout, especially for even larger values?!?

This is now resolved, we have modified to 1 decimal place.

Comment 9. "...damage the user..."? - not traumatize, hurt or injure?

We have modified this text

Comment 10. References are partially correctly listed, therefore, the references: 5, 7, 9-12, 20, 21 should be corrected, the journal/source name is replaced by Elsevier, or even not mentioned at all.

Response. Thank you for spotting this, these are now be correct.

Comment 11. There is no references from the journal MM, although the references are covering recently published papers. It is recommended but not required to cite papers from the journal (this confirms paper compatibility with journal aims & scope, and readers interest in published paper).

Response. On this occasion we did not include a MM paper but we will revise this when submitting our next research.

Comment 12. Dynamical rather than static analysis is recommended for future consideration of ITAP functionality.

Response. This is a good point, the work that we are doing on the ITAP is ongoing with some very interesting findings that we would like to have included in this work. However, for improved clarity we have decided to limit the scope for this paper and consider a follow-up paper in the near future.

Comment 13. Numerical simulations and analyses are great tools, but the true problem of osseointegration and effective prosthetic interventions requires comprehensive understanding of medical devices design and their implementation in living tissue.

Response. This is a very important point. Our contribution is part of ongoing research and there are so many possible avenues for research collaborations, we hope that it leads to more opportunities to work with other research communities in the future.